# Hydrogel-Based Microneedle as a Drug Delivery System

**DOI:** 10.3390/pharmaceutics15102444

**Published:** 2023-10-10

**Authors:** David Filho, Marcelo Guerrero, Manuel Pariguana, Adolfo Marican, Esteban F. Durán-Lara

**Affiliations:** 1Laboratory of Bio & Nano Materials, Drug Delivery and Controlled Release, Department of Microbiology, Faculty of Health Sciences, University of Talca, Talca 3460000, Chile; david.silva@utalca.cl (D.F.); marcelo.guerrero@utalca.cl (M.G.); manuel.pariguana@utalca.cl (M.P.); amarican@utalca.cl (A.M.); 2Center for Nanomedicine, Diagnostic & Drug Development (ND3), University of Talca, Talca 3460000, Chile; 3Institute of Chemistry of Natural Research, University of Talca, Talca 3460000, Chile

**Keywords:** biomedical applications, delivery systems, hydrogels, microneedles

## Abstract

The skin is considered the largest and most accessible organ in the human body, and allows the use of noninvasive and efficient strategies for drug administration, such as the transdermal drug delivery system (TDDS). TDDSs are systems or patches, with the ability and purpose to deliver effective and therapeutic doses of drugs through the skin. Regarding the specific interaction between hydrogels (HG) and microneedles (MNs), we seek to find out how this combination would be applied in the context of drug delivery, and we detail some possible advantages of the methods used. Depending on the components belonging to the HG matrix, we can obtain some essential characteristics that make the combination of hydrogels–microneedles (HG–MNs) very advantageous, such as the response to external stimuli, among others. Based on multiple characteristics provided by HGMNs that are depicted in this work, it is possible to obtain unique properties that include controlled, sustained, and localized drug release, as well as the possibility of a synergistic association between the components of the formulation and the combination of more than one bioactive component. In conclusion, a system based on HG–MNs can offer many advantages in the biomedical field, bringing to light a new technological and safe system for improving the pharmacokinetics and pharmacodynamics of drugs and new treatment perspectives.

## 1. Introduction

The skin is considered the largest and most accessible organ in the human body, and allows the use of noninvasive and efficient strategies for drug administration, such as the transdermal drug delivery system (TDDS). TDDSs are systems or patches, with the ability and purpose to deliver effective and therapeutic doses of drugs through the skin. These strategies have different advantages, mainly in relation to the bioavailability of pharmacologically unstable drugs [1,2]. Microneedles (MNs) are small, pointed projections of micro size adhered or not to support produced by different types of materials. The projections must have well-defined dimensions to allow penetration into the stratum corneum of the skin without pain and without draining blood [3]. Most of these MNs are up to 1500 µm in length, up to 250 µm in width, up to 25 µm in radius, and have a pyramidal and pointed shape (Figure 1) [1,4]. 

Recently, one of the widely studied materials in combination with MNs to create TDDS are hydrogels (HGs). HGs are a three-dimensional network of hydrophilic crosslinked polymers, which can absorb large amounts of water and respond to stimuli. HGs can be made from the crosslinking of monomers belonging to the polymeric chain through chemical interactions (covalent bonds) as well as physical interactions (hydrogen bonds, electrostatics interactions) with crosslinking agents, which will determine some of their mechanical and physical-chemical properties [5].

Depending on the components belonging to the HG matrix, we can obtain some essential characteristics that make the combination of hydrogels-microneedles (HG-MNs) very advantageous, such as the response to external stimuli. pH and temperature are some of the properties inherent to the medium in which HGs can be inserted, and allow molecular modifications in their structure, providing a controlled and sustained release of drugs. Thus, sustained-release local therapy has emerged as a promising alternative to reduce systemic exposure and toxicity, thus increasing the therapeutic outcomes [6].

Interestingly, patches of HG-MNs have not only been used for skin applications. Different studies have reported that these components can be used for specific and deep treatments, giving them a new aspect of utility. For example, Chen et al. (2022) [7] developed a patch based on gelatin methacryloyl HG-MN for the delivery of galunisertib, a transforming growth factor-beta (TGF-β)-specific, for excessive and uncontrollable fibrosis after an attack (myocardial infarction) in vivo. The results were maintained for up to 15 days in a single application and were effective in reducing cardiac fibrosis, protecting from myocardial hypertrophy, and improving cardiac output.

According to Turner et al. (2021) [8], depending on the synthesis process and drug diffusion mechanism, MNs can be subdivided into solid MNs, coated MNs, dissolving MNs, hollow MNs, and forming MNs (Figure 2) with advantages and disadvantages, in which, in some, combination with HG can be used. Solid MNs are formed by materials with great mechanical strength capable of penetrating the skin and leaving micro wounds. After removal, the medicine to be used can be administered. Differently, in coated MNs, the drug load is already adhered to the surface of the MNs so that they penetrate the tissue at the same time as they are released. For dissolving MNs, the bioactive compound and the piercing component are prepared together in a mold. Here, the patch must have sufficient mechanical strength to penetrate the skin and release the therapeutic agent. For hollow MNs, the MNs are hollow with enough space for the drug to reach the tissue. Similarly, in forming MNs, a support has the drug stored which, after penetration by the MNs, can travel around the MNs and reach the tissue.

These materials have a notable advantage over some traditionally occupied drug delivery formulations, mainly with regard to the environmentally responsive capacity of these polymers which, now in combination with hydrogel-based microneedles, provide a standardized system with effective tissue penetration, deposition, and selectivity.

## 2. Hydrogel-Based Microneedle as a Drug Delivery System

As a drug delivery system, hydrogel-based microneedles have several particular characteristics that can be grouped and later compared. In Table 1, there are data related to the type of polymeric hydrogel used, the bioactive component in conjunction with its biomedical research proposal, the type of hydrogel-based microneedles, and their pharmacological responses with regard to controlled and sustained drug release.

## 3. New Approaches for Hydrogel-Based MN Patch 

Among the biomedical applications, we can highlight both the use of hydrogel MNs in systemic and localized therapies, using their various advantages in relation to the controlled, responsive, and induced functionality for drug release (Figure 3). For each biomedical application, important features were found that give the system an advantage over conventional systems. The advantages revolve around: I—stimulus response; II—induced controlled release; III—sustained and long-term therapy; IV—localized and systemic treatment; V—self-activity; and VI—precaution of side effects and low bioavailability; which will be discussed in detail in this chapter based on the results found and disclosed in Table 1. The results discussed in each topic have supporting data available in Appendix A to help the readers identify the articles.

### 3.1. Stimulus Response

Unlike conventional patches, the polymers presented here as HG respond to the environment with which they are associated. In other words, they modify their chemical structure or suffer some molecular rearrangement triggering some change in their elemental properties, depending on the specific characteristics of the associated microenvironment, such as pH, temperature, and reduction–oxidation (REDOX) environment, among others. This is extremely important when referring to application safety and biocompatibility.

Temperature is greatly explored when we talk about HG that responds to stimuli. For nitric oxide (NO) release and biofilm removal in wound healing, a hydrogel composed of PVA HG with S-nitrosoglutathione (GSNO, a NO-releasing agent), and graphene oxide (GO) at freezing temperatures was performed. The test results showed that an increase in temperature increased, through near-infrared irradiation, the release of NO from the patch and thus enhanced antibacterial performance [67].

With distinct applicability using the same principles, a hydrogel based on black phosphorus and thermosensitive pNIPAM was used to control insulin release from a response through near-infrared irradiation. The patch had excellent mechanical properties and demonstrated a controllably released insulin to adjust the blood glucose levels of diabetic mice [110].

In addition, to the ability to respond to stimuli inherent to the environment in which it is inserted, it can store various bioactive compounds to promote dual and/or synergistic therapy. Moreira and collaborators (2020) [15] proposed, through polyvinylpyrrolidone (PVP) MNs coated with chitosan and poly (vinyl alcohol) hydrogel, a dual and targeted therapy with doxorubicin and AuMSS nanorods against cervical cancer cells in vitro. In addition, to having a superior capacity for cytotoxicity to cell lines used, the patch had a temperature and pH response to control the release of bioactive compounds inserted into the matrix.

Enzyme-responsive hydrogel also appears as an attractive alternative to mediate a controlled and sustained release. A protein-based patch of HG–MNs for regenerative internal and external surgical closure was effective for achieving in vitro sustained releases for at least 7 days. In addition to the feat, the capacity of the hydrogel via swelling-mediated diffusion and enzymatic degradation enabled control of the release of the model drug, fluorescein isothiocyanate (FITC)-conjugated dextran [45].

In addition to temperature, enzymes, and other factors as triggers to mediate a patch response to drug release, pH is another characteristic that can be exploited in MN containing HG. Gaware and collaborators (2019) [112] built a patch containing chitosan and graphene assembled in a porous carbon nanocomposites system in addition to responding to an electric field that could respond to pH and release cephalexin. As a result, at acidic pH, the drug was completely released within a period of 48 h, effectively responding to an electric field as well.

Within this pH-controlled release pathway, cancer cells also provide a selective environment for controlled drug release. A study carried out by Cheng and collaborators (2023) [27] using HG–MNs based on methacrylated hyaluronic for delivery of acid estrogen receptor alpha (ERα)-degrading PROTAC-ERD308 and palbociclib within micelles responsive to tumor acid pH was developed for effective treatment of ER-positive breast cancer. The synergism sought was highly selective with retentive properties for neoplastic cells.

As we have seen, there are several external stimuli that can trigger a response for controlled and sustained release of therapeutic agents from a system of HG–MNs. Exploring these specific characteristics of the environment in which it is inserted, we can use them to generate an even more selective system for the therapeutic objective, thus preventing systemic adverse effects and enabling even more selective responses to the tissue or region of interest.

### 3.2. Induced Controlled Release

One of the main advantages of using HG to make patches for treatment is the possibility of inducing highly controlled drug release. Much has been said about the response to stimuli inherent in the environment in which these smart polymers are found, but few studies explore the possibility of other external stimuli (parts in the environment) to modulate the release of the bioactive compound. External stimuli such as electricity are well explored regarding hydrogel MN and the application of localized and often systemic therapy.

Using an ex vivo porcine model, an HG–MNs composed of a mixture of 6% PVA, sodium hydroxide, poly (ethyleneimine), and 1-vinylimidazole, and N, N′-Methylenebisacrylamide (crosslinking agent) was able to release indomethacin in a highly controlled manner through electrical stimuli. Ex vivo evaluation of the EMH–MNA device across porcine skin demonstrated that without electrostimulation, significantly less drug release was obtained (±0.4540 mg) as compared to electrostimulation (±2.93 mg) [78]. The same authors, however, now in vivo and in vitro models, also proved the effectiveness of the patch for electrostimulated controlled release [84].

Similarly, another study highlighted in Table 1 demonstrated similar results with electricity-responsive patches for drug delivery. Jeong and collaborators (2023) [95] constructed patches, now composed of polyvinylpyrrolidone-polysaccharide (PVPS)-based graphene oxide (GO) HG, for the release of L-ascorbic acid. The drug-release efficiency of the PVPS/GO microneedle patch was increased by more than two times compared to the PVPS microneedle patch (without applying electricity) when 5 V was applied (Figure 4).

Other external stimuli can be exploited and combined to release drugs from these patches containing HGMNs. Yang and coworkers (2021) [98] made a patch of PVP in which the back was covered with gold and silver to make it respond to electricity when a current was applied, and to temperature. When irradiated by infrared radiation, the drug-release efficiency increased more than 7.9 times compared to a flat PVP patch (without applying electricity) when 3 V was applied. In addition, when 40 °C heat was applied, the drug-release efficiency of the Au- and Ag-coated PVP based on microneedle patches improved about 5.3 times compared to a flat PVP patch at room temperature.

Perhaps in the future, with the advancement of these technologies, we will find hydrogel-based patches that can deliver the drug or mediate its action as needed. In other words, that release will occur when we want it or when we need it. The studies of these systems and their applicability involving the generation of voluntary external responses are extremely interesting when we work with the ideas of HG-MNs and their biomedical applications.

### 3.3. Sustained and Long-Term Therapy

It is well known that HG can mediate a long-term response. Another advantage associated with the combination of MNs and HG is the possibility of retaining the bioactive compound for prolonged release. The factors related to this depend directly on the hydrogel formulation, in terms of its composition. For example, a hydrogel made from gelatin crosslinked with metacryloyl using UV as initiator allowed controllable doxorubicin release rates from the degree of the crosslinking polymer. Furthermore, the mechanical properties of the component could also be adjusted and improved with this factor [10].

Comparing the HG–MNs system with other forms of release, such as the passive diffusion route, we can visualize some other interesting advantages. When compared, HG–MNs based on PVA hydrogel were extremely efficient for in vitro release of doxorubicin when compared to passive diffusion of the drug and proven antineoplastic effect [17]. Zhou and coworkers (2020) [9] also compared their patches with a nontransdermal treatment system, in which the patch containing β-Cyclodextrin conjugated methacryloyl exhibited relatively higher therapeutic efficacy than curcumin through a more localized and deeper penetrated manner compared with a control nontransdermal patch.

Interestingly, while an attack concentration of the drug can be reached in the first hours of use, a gradual diffusion can be achieved, and a few days of sustained release can be achieved in addition. Darge and collaborators (2022) [20] achieved an adequate release of DOX and lipopolysaccharides in the first 24 h (66.1 ± 7.4% and 59.4 ± 5.5% of DOX and LPS, respectively), followed by a gradual and sustained release of the drugs that lasted around 7 days. Pillai and coworkers (2023) [100], with a PVA-based patch, also achieved 12-day release using rhodamine as the model drug.

Extremely specific deep treatments can benefit from using these patches with the combination of HG-MNs. Fibrosis after myocardial infarction (MI) in the peri-infarct zone leads to left ventricular remodeling and deterioration of cardiac function. An HG–MNs patch composed of gelatin methacryloyl hydrogel microneedle for delivery of galunisertib, a TGF-b (fibroblast activator) inhibitor, was extremely efficient for inhibition of TGF-β/Smad2 signaling, contributing to reduced cardiac fibrosis, protecting from myocardial hypertrophy, and improving cardiac output with a sustained release of up to 15 days after a single application [7].

However, a prolonged release of the drug is not always sought, and this system is still effective for such a strategy. In a study carried out by Alafnan and collaborators (2022) [25], thinking about a rapid release system, 100% of the gemcitabine was released in the first hour of administration with a system manufactured with PEGDA, gaining high retention in the tumor region. This exemplifies the versatility of these materials not only for prolonged release over time but also for effective penetration and retention in the intended locations without affecting other tissues.

It is not only necessary that the drug be released steadily, but also that the impacts are sufficiently achieved. Using Prussian blue nanozymes (PBNs) and vascular endothelial growth factor (VEGF) encapsulated in multifunctional silk fibroin methacryloyl microneedle, for wound closure healing, the effects of the combination of components were extremely beneficial, with a lasting effect for up to 9 days of treatment evaluated in vivo through antioxidative stress and antiapoptotic mechanisms [54]. Similarly, now using Ti_2_C_3_ MXenes-integrated poly-γ-glutamic acid (γ-PGA) hydrogel MNs to deliver asiaticoside in diabetic wound healing, Wang and coworkers (2022) [55] achieved a lasting effect of up to 14 days with the combination with MNs. A 14-day prolonged effect for systemic parathyroid hormone release and promotion of collagen deposition during skin healing with PVA-based HG–MNs was also achieved by Yao et al. (2021) (Figure 5) [41].

On a therapeutic level, peptides have attracted great attention in recent years due to their wide applicability. HG-MNs are compatible with delivering antibodies for the most diverse areas of biomedical applications. In this broad sense, a study carried out by Courtenay and coworkers (2018) [18] for the delivery of bevacizumab and treatment of lymphoma using this technology allowed a release for more than 7 days of the bioactive compound from a single application. Furthermore, in vivo studies detected the peptide in lymph nodes, spleen, and cutaneous tissues, indicating accumulation in the lymphatic system, and an interesting applicability for sustained delivery and treatment of lymphomas as well as metastatic tumors.

As stated earlier, HG for sustained drug release is not unheard of; what is new is the advantage that these elements bring with their use and combination with MNs. In this sense, therapies that require a prolonged effect with the possibility of self-application and independence should benefit from these noninvasive systems.

### 3.4. Localized and Systemic Treatment

Both approaches can be achieved when talking about MN–HGs, depending on the objective and the disorder to be treated. In the case of melanomas, for example, localized therapy is required to avoid systemic exposure to the drug used. On the other hand, to control metabolic disorders, for example, in which there is a need for systemic control of glucose concentrations, therapy with the application of these patches is also suitable for the proposal. The most important here is the achievement of concentrations considered therapeutic without the need for replacement, or few replacements throughout a treatment.

Based on therapeutic concentrations of esketamine in the treatment of depression, Courtenay and coworkers (2020) [58] proposed a patch of HG–MNs based on poly (methyl vinyl ether/maleic anhydride) (Gantrez^®^ S-97). The system achieved sustained therapeutic levels of 0.15–0.3 μg/mL in plasma over 24 h using ESK-containing drug reservoirs in combination with hydrogel-forming MNs in this in vivo feasibility study. Kearney et al. (2016) [59] also reached therapeutic concentrations (854.71 µg ± 122.71 µg) over 24 h for Donepezil, using a patch with the same polymeric system (Gantrez^®^ S-97).

HG-MNs systems have been widely used to treat metabolic disorders, especially diabetes, involving systemic therapies. As an example, a patch containing poly-c-glutamic acid (c-PGA) MNs and PVA/PVP hydrogel developed by Chen et al. (2015) [49] enabled the delivery of insulin for the treatment of diabetes, without requiring the users to remove any sharps or waste as the main advantage. These results suggest that the MN system featuring this unique design enables a quick convenient self-administration method.

HG in this combination has also been studied to deliver compounds capable of promoting vaccine-based immune responses. With the pandemic eminence associated with coronavirus (SARS-CoV-2), Kim and colleagues (2020) [28] developed a patch of HG–MNs based on carboxymethyl cellulose for the delivery of protein MERS-S1f, MERS-S1fRS09, MERS-S1ffliC, SARS-CoV-2-S1, or SARS-CoV-2-S1fRS09 for antigen stimulator. In good response, the combination generated strong and long-lasting antigen-specific antibody responses. Similarly, aiming at self-vaccination and increased immunogenicity via direct targeting of skin dendritic cells, Courtenay and col (2018) [29] encapsulated protein antigen ovalbumin (OVA) within Gantrez^®^ S-97 and poly (ethylene glycol) (PEG) for controlled release. In this study, the system provoked responses to produce antibodies considerably around the 70th day of evaluation in a booster strategy at 14, 42, and 70 days. Dissolving MN arrays have yielded enhanced immune responses, suggesting the possibility of lower dosing and equivalent outcomes as traditional needle and syringe methods (Figure 6).

The treatments of disorders involving the male reproductive system, such as erectile dysfunction, also generate some problems with traditional therapies and can benefit from the use of patches containing HG–MNs, mainly regarding prolonged therapy and self-administration. An ex vivo permeation study, using PVA and PVP as polymers and several variations of tartaric acid as crosslinking agents, showed that up to 80% of sildenafil citrate (equivalent to 20.2 ± 0.29 mg/mL) was delivered transdermally [76]. Using the same sildenafil citrate and with the same polymeric composition, but with citric acid as a crosslinking agent, the same investigation group (Fitri et al. (2023) [93]), now aiming at an application on pulmonary hypertension, obtained good results through in vitro studies to suppose the good patch applicability for this function. In these studies, they demonstrated, for the first time, the possibility of a system of HG–MNs to deliver sildenafil citrate aiming at administration of sildenafil citrate and prevention of problems of low oral bioavailability.

The range of applicability of these components is extremely versatile. Given all the elements and characteristics seen so far, most of the results lead us to think that these systems applied to the skin would be limited to that. The possibility of systemic therapies also emerges as another important feature of these elements, serving as a new gateway to advanced treatment proposals.

### 3.5. Self-Activity

Another great advantage of the patch containing HG is the activity that these elements possess. Self-activity here is referred to as an intrinsic pharmacological property of the matrix, depending on the polymeric components. In this specific case, in addition to keeping other bioactive substances in storage, the network itself could have a dynamic effect.

Chitosan is a natural polysaccharide commonly used to make these smart polymer systems because of its high biocompatibility, drug-retentive capacity, and antimicrobial properties [72]. In addition to this outstanding ability, chitosan can be combined with other monomers with polymeric capacity to further increase its range of applicability. For the delivery of vascular endothelial growth factor (VEGF), a patch (type of HG–MN) with the combination of chitosan and pNIPAM, was extremely efficient in encapsulating large amounts of the drug and promoting acceleration in inflammatory inhibition, collagen deposition, angiogenesis, and granulation tissue formation in vivo. In addition, to the effectiveness for wound healing for 9 days using VEGF, the superstructure responded to temperature and exerted antimicrobial effects, tested on strains of *S. aureus* and *E. coli* in vitro (Figure 7) [70].

A synergistic effect can also be visualized with the use of these components present in the superstructure in the patches format of HG–MNs. To eradicate biofilm formation in vitro, a patch of HG–MNs containing chitosan and zinc nitrate (Zn^2+^) was tested and compared to a membrane lacking needles. The structure was extremely superior in all aspects evaluated, with complete inhibition of antimicrobial growth and excellent antibiofilm activities, in addition to good biocompatibility evaluated by 3-(4,5-dimethylthiazol-2-yl)-2,5-diphenyltetrazolium bromide (MTT) assay. These data emphasize that the way the drug is delivered to the environment in which it is inserted is extremely important for the efficiency of the components of the structure [71].

Chitosan’s extended property is extremely flashy. In addition to exerting efficient effects on different types of bacterial cells, the antifungal property can also be interesting using applicability through HG–MNs. The synergistic effect has now been sought in combination with Amphotericin B to eradicate *C. albicans* formation in vivo. The effectiveness of the structure was associated with the high bioavailability of therapeutics and synergistic actions of the antifungal polymer and drug [72].

The possibility of effectiveness without exposing concentrations of exogenous components to the body, promoting an even more localized and safe therapy, is the main highlight of this topic. Based on this evidence, these elements can be conjugated to MNs to promote the retention of the system in an even more specific environment and safely exert their effects apart from the presence of other additional substances.

### 3.6. Precaution of Side Effects and Low Bioavailable

Another advantage of the patch itself, which does not directly depend on the hydrogel formulation, is the possibility of precaution of the adverse effects and increasing the bioavailability of some drugs. One of the major problems of insulin-dependent patients is hypoglycemia. An HG–MN manufactured with photocrosslinked methacrylated hyaluronic acid (MeHA) that was glucose-responsive was designed to respond to low glucose concentrations and mediate glucagon release. Incredibly, the patch was effective in preventing an insulin overdose and consequent hypoglycemia [52].

In a similar vein, now applied to the treatment of opioid-induced pruritus, an HG–MNs patch composed of polyacrylic acid was used to deliver naloxone. In addition to making it possible to manage the effects caused by the use of opioids, the hydrogel was able to respond to changes in pH. Naloxone permeation through intact skin was highest from pH 7.4 gels when naloxone was unionized, in contrast to undetectable concentrations permeated from pH 5 gels with 100% ionization. Combining MN treatment with pH 5 gels significantly enhanced permeation and resulted in a steady-state flux that would achieve therapeutic delivery (Figure 8) [92].

L-DOPA in its traditional route of administration, the oral route, allows major gastrointestinal adverse effects. In the treatment of a Parkinson’s mice model, this regimen significantly restored motor function, and further mechanistic studies revealed that it reduced neuroinflammation and dopaminergic neuronal death in the substantia nigra. In addition, the released L-DOPA directly enters the blood, which reduces the side effects on the gastrointestinal tract and improves the utilization rate of the drug [60].

Although this feature does not depend directly on the combination of both elements, being essentially due to the presence of a format in MNs, it is still part of the system and is described here as an advantage over other routes of administration. The benefit of combining these elements, dealt with in depth in the other topics, can be added to this to further enhance the range of applicability of these components.

## 4. Biomedical Applications of Hydrogel-Based Microneedles

According to the analysis from the general data extracted from Table 1, materials based on HG in MNs have different applicability and are distributed in the most diverse local (on-site) and systemic (remote control) biomedical areas (Figure 9). Central nervous system (CNS) disorders such as depression, Parkinson’s, and Alzheimer’s, as well as problems related to both male and female reproductive systems, are some of the specific proposals for why these systems were effective. Based on this, each element was classified in its general application according to the system in which it was proposed.

Our focus in this topic was on verifying which are the main areas of application of HG–MNs of these new systems for the controlled, localized, and sustained release of drugs to later group them into statistical values.

### 4.1. General Characteristics for Area in Biomedical Application 

For the studies carried out, each of the results was reallocated into groups according to their purpose and biomedical applicability (Figure 10). The results were classified into cancer treatment; vaccinations, autoimmune disorders; wound healing; metabolic disorders; diseases of the central nervous system (CNS); eye disorders; infectious disorders; disorders of the reproductive system; analgesic, anesthetic and/or anti-inflammatory effect; skin lightening and skin antiaging agent; tissue regeneration; management of pharmacokinetic and toxicity parameters; pulmonary disorder; and not specified for those that did not specifically refer to which pathology they could be applied. Statistical aspects were traced in relation to the results in order to highlight the areas of possible applicability involving hydrogel MNs.

Among all analyzed articles, there is a prevalence of 18.18%, 11.81%, and 11.81% of HG–MNs being applied in cancer, metabolic disorders, and infection disorders, respectively. Disorders, including cancers and skin infections, that develop in this area of the skin are widely applicable to these patches, being the main explanation as to why this factor has been so exploited. For metabolic disorders, specifically in relation to diabetes, there is a great need to promote better dosage management of application and adverse effects resulting from insulin treatment. In addition, bringing a new form of intelligent and responsive systems independent of human needs is directly applicable to this pathology.

A new presentation for the application of these components gives the possibility of a new treatment system with greater effectiveness, where the penetration of the skin and treatment in deep tissues is one of the greatest advantages in relation to the commonly used routes of administration. In general, we found different applicability ranging from localized to systemic treatment, proving to be a new modality to intervene in different types of pathologies.

### 4.2. General Characteristics for Hydrogel Type of HG–MNs

Soon after the general classification of the elements, they were submitted to a new classification according to their type (Figure 11) The results were classified into hydrogel forming MN; hydrogel dissolving MN; hydrogel solid MN; hydrogel hollow MN; and hydrogel coated MN.

As expected, hydrogel dissolving MNs (68.18%) was the main type of hydrogel developed in this analysis. Comparing this to other types of HG in combination with MNs, some studies revealed that this one is efficiently better, being able to deliver better amounts of drug and having a better release profile. This would be the main justification for why this is more prevalently studied in the literature compared to other types of patches and release kinetic mechanisms. 

For example, Aung and coworkers (2020) [87] compared three systems for delivering alpha-arbutin, a skin lightener: (a) forming MNs, (b) dissolving MNs, and (c) commercial cream. Interestingly, dissolving MNs was superior in both in vivo and in vitro studies to the other two, with high permeability capability and high transdermal delivery of alpha-arbutin. Another study by Courtenay and coworkers (2018) [18] found similar results for the two presentations of HG–MNs, which were effective for bevacizumab delivery and antineoplastic treatment with interesting data, discussed earlier.

In every sense, the presentations of HG–MNs bring a new perspective of interaction between active compounds and the human body, enabling the application of the latest in drug delivery systems, the combination of MNs and HG.

## 5. Drawbacks and Future Perspectives

Observing the most diverse applicable biomedical areas and combinatorial advantages of the use of hydrogel-based microneedles, the system in recent years is gaining more and more attention. For instance, the DP4 microneedling device (K221070), INTRAcel RF Microneedle System (K183284), and MICRONJET 600 (K092746) are some systems approved by the Food and Drug Administration (FDA) with different purposes that exemplify the great ease and safety of these materials in terms of their use and effectiveness [114].

However, some disadvantages can be observed within the reviewed elements. One of these problems is related to mechanical strength, in which not all polymeric materials can be used. During the review, and as can be seen in Table 1, there was a considerable prevalence of PVA or PVP in the formulation of the systems; this also brings possible limitations to their advantages (highlighted during the discussion), being overcome when used in conjunction with other elements. Thus, there will necessarily be an element with considerable mechanical strength during the formulation of these systems [76,82,83,84,93,98,99,100].

Some of these elements also showed rapid drug loss, indicating possible retention failures. This depends not only on the polymeric system used and the drug but also on the type of HG-MNs manufactured. This report was seen in a few studies, mainly evidenced when two types of HG-MNs were presented in a comparative way. In most of the elements mentioned, the systems were precise and revealed a long retention and maintenance time for the retained drug, releasing it over the course of days, weeks, and even months [14,18,57,67]. Nevertheless, this cannot even be considered a disadvantage when we think about a system of fast effective penetration and high localized release [27].

These elements, despite some controllable and circumventable limitations, bring to light a compact, safe, and efficient system for administering drugs in their most diverse mechanisms of action and perspectives, proving effective to be combined with the latest generation technologies, such as gene silencing for the treatment of cancer, for example [26]. Taking into account all these elements that bring disadvantages and advantages of this combination, HG–MNs should continue to be the basis for exploring more innovative and technological systems for this modality of drug release and treatment.

## 6. Conclusions

Throughout this focused review of the recent developments in HG-MNs, we discussed the applications in biomedical science as they pertain to drug delivery and controlled and sustained release. Cancer (18.18%), metabolic disorders (11.81%), and infection disorders (11.81%) were the prevalent biomedical applications during this analysis, and the prevalent type of hydrogel MNs was hydrogel dissolving MNs, with a percentage equivalent to (68.18%), which reveals greater kinetic efficiency of this patch compared to the other formulations used. Some of the systems were compared with the traditional methods of administration, where they had some advantages exhibiting biocompatibility and efficiency for which they were proposed. Moreover, aspects related to (1) induced controlled release; (2) stimulus response; (3) sustained and long-term therapy; (4) localized and systemic therapies; (5) self-activity; and (6) precaution of side effects and low bioavailability were selected based on the analysis as advantages of the combination of HG and MNs. The great applicability of these components, regarding the terms mentioned above, accredits them to a new aspect of drug delivery systems for localized and systemic treatment. Controlled, sustained, and localized release, as well as the possibility of a synergistic association between the components of the formulation (charged bioactive components and some polymeric systems), and the combination of more than one bioactive component in a single system, can be used as an advantage and principle for the exploration of more systems, bringing to light a new technological and safe system for improving the pharmacokinetics and pharmacodynamics of drugs and new treatment perspectives.

## Figures and Tables

**Figure 1 pharmaceutics-15-02444-f001:**
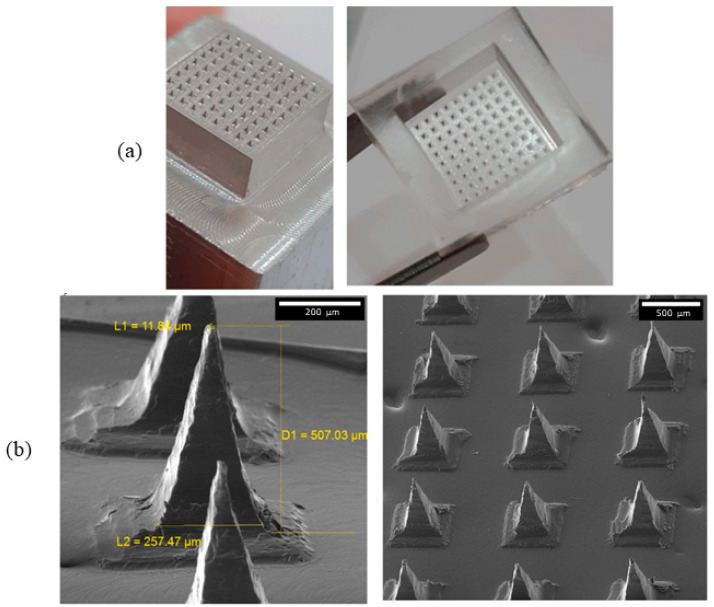
Dissolving MNs patch dimensions and shape [4]. (**a**) Dimensions and shape; (**b**) scanning electron micrographs (SEM) of the dissolving MNs. Reproduced under terms of the CC-BY license [4]. Copyright 2023, Malek-Khatabi et al., published by Elsevier.

**Figure 2 pharmaceutics-15-02444-f002:**
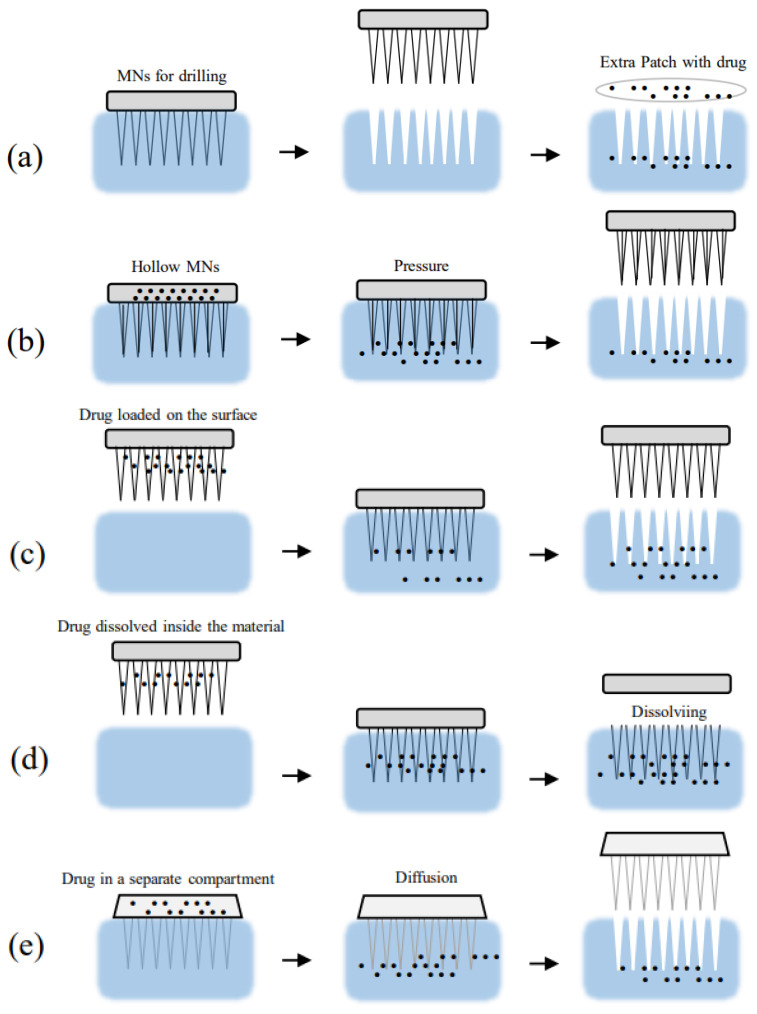
MNs classification according to the synthesis process and diffusion mechanics. (**a**) Solid MN, (**b**) hollow MN, (**c**) coated MN, (**d**) dissolving MN, and (**e**) forming MN.

**Figure 3 pharmaceutics-15-02444-f003:**
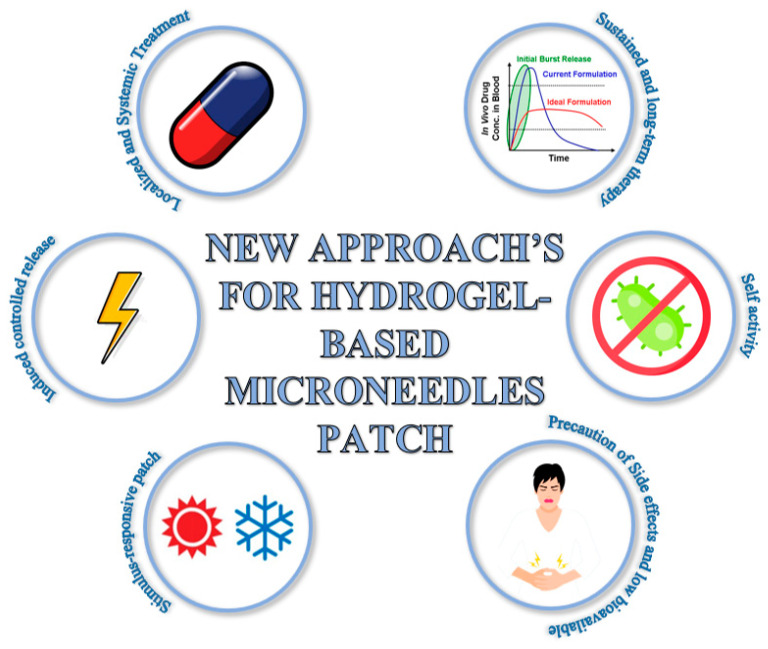
New approaches for hydrogel-based MN patch.

**Figure 4 pharmaceutics-15-02444-f004:**
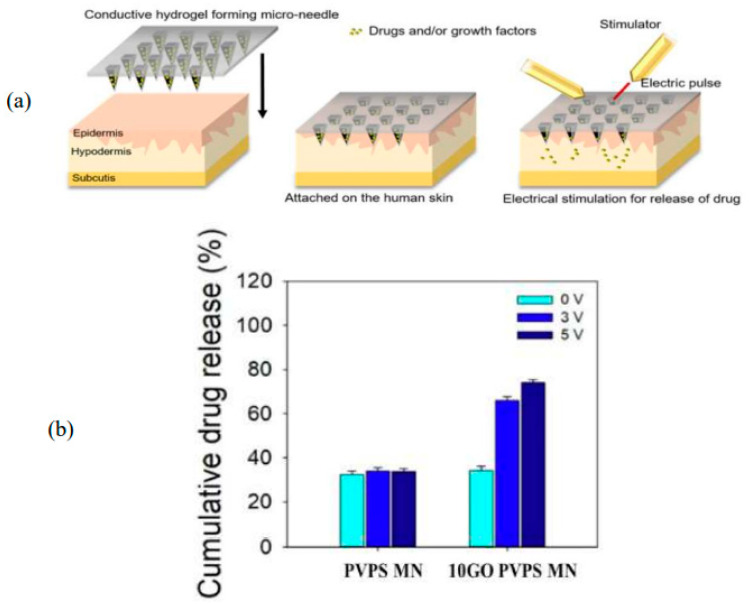
PVPS/GO electro responsive HG–MNs [95]. (**a**) Schematic drawing of patch and stimulation mechanism; (**b**) cumulative drug release with electric stimulus and without stimulus in PVPS with GO and without GO. Reproduced under terms of the CC-BY license [95]. Copyright 2023, Jeong et al., published by Elsevier.

**Figure 5 pharmaceutics-15-02444-f005:**
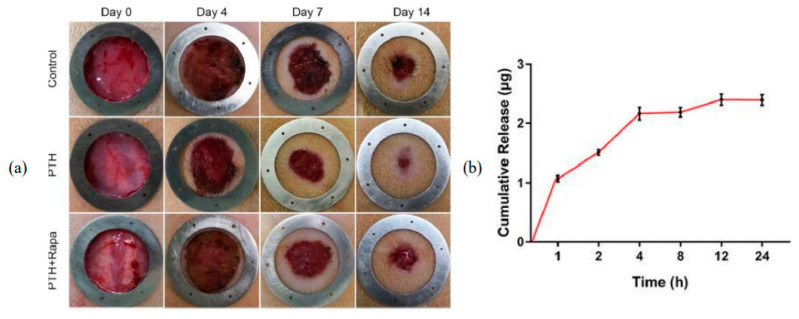
Wound healing and cumulative release profile of parathyroid hormone in PVA-based HG–MNs [41]. (**a**) Wound healing profile in 14 days; (**b**) cumulative release of parathyroid hormone. Reproduced under terms of the CC-BY license [41]. Copyright 2021, Yao et al., published by Elsevier.

**Figure 6 pharmaceutics-15-02444-f006:**
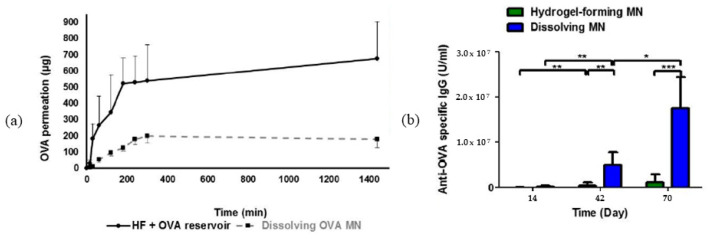
Permeation of OVA through HG forming/dissolving MN and antigenic-specific responses to therapy [29]. (**a**) OVA permeation across excised neonatal porcine skin over 24 h; (**b**) responses for IgG production with booster strategies at 14, 42, and 70 days (where (*) is *p* < 0.05, (**) is *p* < 0.01 and (***) is *p* < 0.001—Means ± S.D, n = 8). Reproduced under terms of the CC-BY license [29]. Copyright 2018, Courtenay et al., published by American Chemical Society (ACS).

**Figure 7 pharmaceutics-15-02444-f007:**
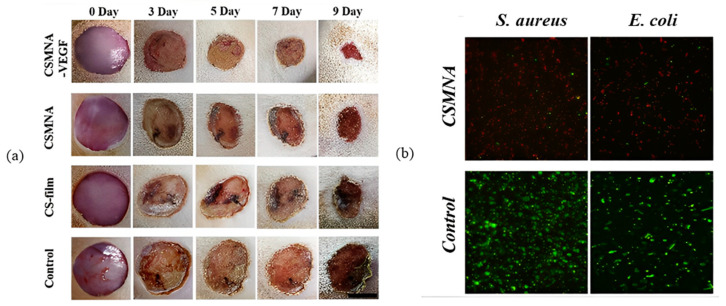
Wound healing and antibacterial effect of chitosan HG–MNs patch and VEGF combination [70]. (**a**) Wound healing profile; (**b**) antibacterial effect on *S. aureus* and *E. coli*. Reproduced under terms of the CC-BY license [70]. Copyright 2020, Chi et al., published by Keai Publishing.

**Figure 8 pharmaceutics-15-02444-f008:**
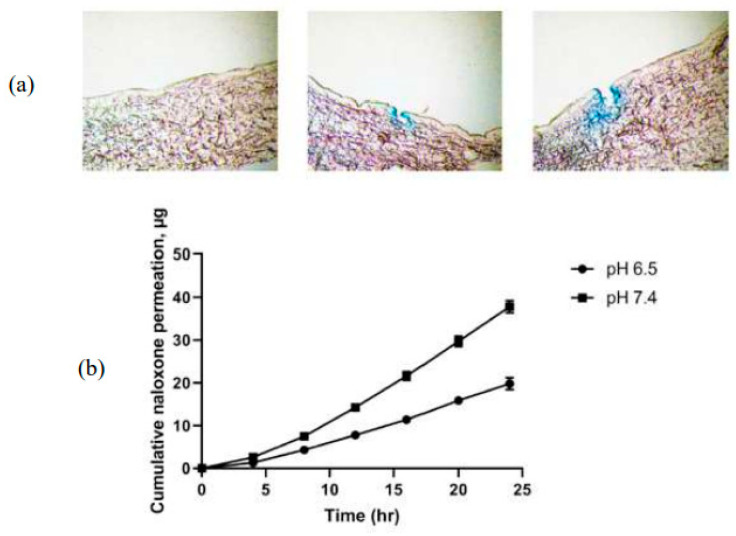
Cyrosectional plot of intact and lesioned porcine ear skin and cumulative naloxone permeation of polyacrylic acid HG-MNs on different pHs [92]. (**a**) Cyrosectional plot of intact and lesioned porcine ear skin with MNs of 500 μm and 750 μm, respectively; (**b**) the cumulative permeation of naloxone. Reproduced under terms of the CC-BY license [92] Copyright 2019, Gao et al., published by Elsevier.

**Figure 9 pharmaceutics-15-02444-f009:**
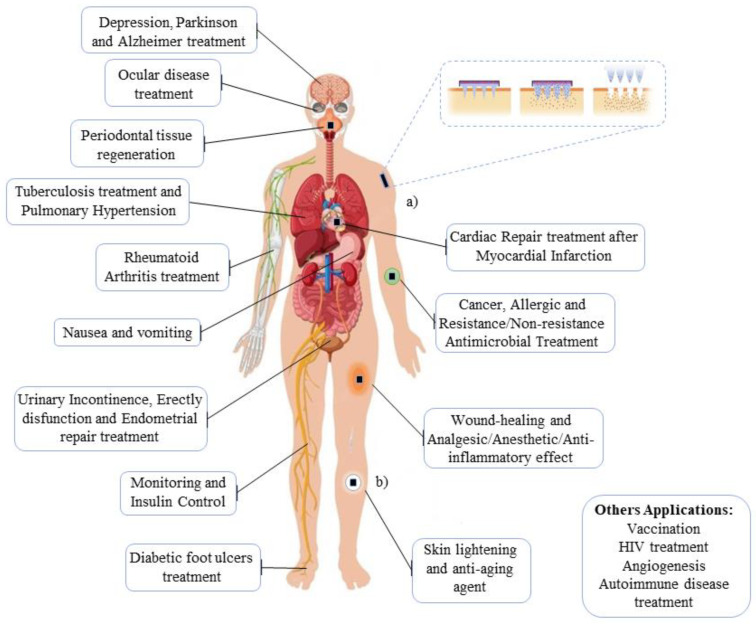
The results of hydrogel microneedles applied to specific biomedical applications. (**a**) Systemic treatment: HG–MNs can be used for systemic disorders allowing remote control. (**b**) Local treatment: in another sense, the system can be used in situ, onsite.

**Figure 10 pharmaceutics-15-02444-f010:**
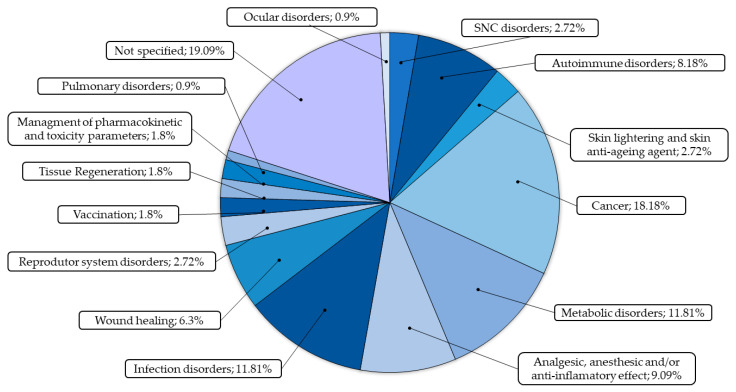
Percentual graphic of the application of hydrogel MN general areas.

**Figure 11 pharmaceutics-15-02444-f011:**
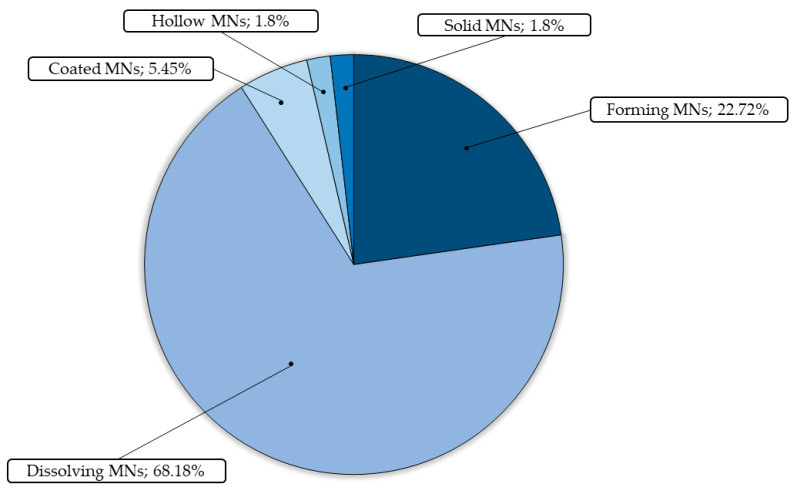
Percentual graphic of the type of hydrogel MNs in biomedical applications. During the analysis of the samples that contained more than one type of MN under test, the count value was assigned to the element with the highest percentage proportion in the final count.

**Table 1 pharmaceutics-15-02444-t001:** Results according to their application, composition, bioactive compound, type, and release characteristics for hydrogel system.

Hydrogel System	Bioactive Compound and Defined Specific Proposal (DSP)	Type	Response and Release Characteristics	Ref.
CANCER TREATMENT				
β-Cyclodextrin conjugated gelatin methacryloyl	Curcumin DSP: Melanoma	Dissolving MNs	The gelatin methacryloyl (GelMA)-β-CD/CUR MN exhibits relatively higher therapeutic efficacy through a more localized and deeper penetrated manner compared with a control nontransdermal patch for anticancer activities. In vivo, studies also verify the biocompatibility and degradability of the gelatin methacryloyl (GelMA)-β-CD MN arrays patch.	[9]
Gelatin methacryloyl (GelMA)	Doxorrubicin DSP: Melanoma	Dissolving MNs	The drug release rate can be adjusted by controlling the degree of polymer crosslinking.	[10]
Human serum albumin methacryloyl hydrogel	Doxorubicin DSP: Melanoma	Dissolving MNs	DOX-embedded human serum albumin methacryloyl displayed good anticancer efficacy with sustained activity up to 24 h.	[11]
Poly (ethylene glycol) diacrylate (PEGDA) hydrogel	Doxorubicin and capecitabine DSP: Mammary carcinome	Coated MNs	They used a new system to monitor the amounts of drugs release.	[12]
Hyaluronic acid dissolving MNs (HA DMNs)	Doxorubicin DSP: Skin cancer	Dissolving MNs	Two systems were used: swelling MNs and dissolving MNs. The relative bioavailability of Doxorubicin in swelling MNs towards Doxorubicin in dissolving MNs was 200% within 12 h.	[13]
Dextran methacrylate hydrogel	Doxorubicin and trametinib DSP: Melanoma	Dissolving MNs	The prepared hydrogel MNs can successfully penetrate the epidermal layer and achieve sustained drug release. Tested for 12 days.	[14]
Polyvinylpyrrolidone MNs coated with chitosan and poly (vinyl alcohol) hydrogel	Doxorubicin and AuMSS nanorods (Dox@MicroN) DSP: Cervical cancer	Coated MN	MNs structures can efficiently penetrate the tumor-mimicking agarose gel and release the Dox with a pH- and thermo-responsive profile. Furthermore, the system composed of Dox and AuMSS nanorods was able to simultaneously mediate the chemo- and photothermal therapies rendered a superior cytotoxic effect against the cervical cancer cells.	[15]
Polyvinylpyrollidone-co-vinyl acetate	Imiquimod DSP: Basal cell carcinoma (BBC)	Dissolving MNs	Permeation studies utilizing Franz diffusion cells demonstrated that the imiquimod-loaded polymeric MNs were capable of delivering similar quantities of imiquimod to the region of tumors, despite a six fold lower drug loading, relative to the current clinical dose of Aldara^TM^ cream used in BCC treatment.	[16]
Poly (vinyl alcohol)	Doxorubicin DSP: Skin cancer	Forming MNs	The insertion of MNs resulted in significantly greater delivery of doxorubicin into and across human skin, as compared to passive diffusion.	[17]
Poly (vinyl alcohol) and poly (methyl vinyl ether co-maleic anhydride/acid (Gantrez^®^ S-97), polyethylene glycol (PEG 10,000), and Na_2_CO_3_	Bevacizumab DSP: Immunotherapy for cancer	Dissolving and forming MN	BEV was detected and measured in plasma across 7 days following one single application of MN arrays. BEV had lymphatic accumulation in vivo. This could prove to be a viable option for the treatment of lymphomas and secondary metastatic tumors.	[18]
Carbopol hydrogel	Epigallocatechin-3-gallate DSP: Skin cancer	Dissolving MNs	Microneedle-treated skin showed significant enhancement in the delivery of EGCG to viable epidermis and dermis from the hydrogel in comparison with the untreated skin.	[19]
Sodium alginate and sulfobetaine methacrylate using N, N′-methylenebisacrylamide and Ca^2+^	Doxorrubicin and lipopolysaccharides DSP: Glioma tumor	Dissolving MNs	Adequate amounts of drugs were released from the MNs in the first 24 h, followed by a very gradual and sustained release for the next 7 d.	[20]
Protoporphyrin (PpIX) and dihydroartemisinin (DHA) with hyaluronic acid	Dihydroartemisinin DSP: Melanoma	Dissolving MN	The system was pH-sensitive and efficient to promote photodynamic therapy under light irradiation and amplified generation of reactive oxygen species.	[21]
Sodium hyaluronate (HA)	Palonosetron hydrochloride DSP: Skin cancer	Dissolving MN	The system showed rapid dissolution in the skin and there were no statistically significant differences in Tmax and AUC when compared to subcutaneous administration.	[22]
Gelatin methacryloyl (GelMA)	Gemcitabine DSP: Pancreatic cancer	Dissolving MNs	In vitro experiments demonstrated the outstanding adhesive ability on irregular surfaces in moist environments, and the drug-releasing kinetics were elucidated to be well controlled by adjusting the concentration of gelatin methacryloyl (GelMA).	[23]
Poly (methylvinylether/maelic acid) and crosslinked with glycerol hydrogel	5-aminolevulinic acid (ALA) or meso-tetra (N-methyl-4-pyridyl) morphine tetra tosylate (TMP) DSP: Photodynamic therapy for cancer	Dissolving and forming MNs	TMP delivery was much more marked from the hydrogel-forming system, which offers another advantage over its dissolving counterpart in that the MN is removed intact, thus leaving no polymer behind in the skin.	[2]
Gantrez^®^ S-97 hydrogel and polyethylene glycol (PEG)	Gold nanorods DSP: Basal cell carcinoma	Forming MNs	In the formulation, the heat generated rapidly by the combination allowed the temperature to reach 50 °C in a few seconds. Compared to conventional methods, delivering the component in a minimally invasive way and without leaving polymeric and particulate material on the skin are the main advantages.	[24]
Polyethylene glycol diacrylate (PEGDA)	Gencitabine DSP: Inflammatory breast cancer	Coated MNs	Thinking about a rapid release system, 100% of the drug was released in the first hour of administration with a system manufactured with high retention.	[25]
Silicon	Cholesterol-modified housekeeping gene (Gapdh) siRNA DSP: Skin cancer and other skin disorders	Solid MNs	The strategy reduced the expression of the gene of interest by up to 66% specifically in the skin without affecting other organs.	[26]
Methacrylated hyaluronic acid	Estrogen receptor alpha (ERα)-degrading PROTAC—ERD308 and Palbociclib DSP: ER-positive breast cancer	Dissolving MNs	The patch was effective in overcoming the problems involved with PROTAC systems, which had high rates of local drug retention (87%). Furthermore, the combination with pH-responsive micelles allowed the co-release of Palbociclib specific to the tumor acid environment for effective therapy against breast cancer.	[27]
VACCINATION				
Carboxymethyl cellulose	Protein MERS-S1f, MERS-S1fRS09, MERS-S1ffliC, SARS-CoV-2-S1, or SARS-CoV-2- S1fRS09 for antigen stimulator DSP: Coronavirus (SARS-CoV-2)	Dissolving MNs	The hydrogel-based vaccine MNs elicited strong and long-lasting antigen-specific antibody responses.	[28]
Gantrez^®^ S-97 and poly (ethylene glycol) (PEG)	Protein antigen ovalbumin (OVA) DSP: Adjuvant for vaccination	Forming and dissolving MN	The system yielded enhanced immune responses, suggesting the possibility of lower dosing and equivalent outcomes as traditional needle and syringe methods.	[29]
AUTOIMMUNE DISORDERS				
Poly (vinyl alcohol)/poly vinyl pyrrolidone	Methotrexate DSP: Psoriasis	Dissolving MNs	The system acted as a drug depot and released the MTX in a sustained manner over 72 h, while minimizing MTX systemic exposure. Indeed, 24 h after application, a concentration of 322-fold higher than the amount of MTX retained in the skin after oral administration of MTX Na was reached.	[30]
Dextran methacrylate and cyclodextrin-adamantane	α-MSH and tofacitinib DSP: Vitiligo	Dissolving MNs	Under the treatment of α-MSH/tofacitinib MNs, massive deposition of melanin in epidermis and hair follicles significantly accelerated skin and hair pigmentation.	[31]
Gantrez^®^ S-97, polyethylene glycol and poly (lactic acid)	Tofacitinib citrate DSP: Vitiligo, psoriasis, etc.	Hollow and dissolving MN	The dissolving MN arrays showed superiority in this regard, as around 835 μg/cm^2^ were found on the dermis after 24 h of study, providing proof of principle for intradermal delivery of tofacitinib citrate using MN arrays.	[32]
Hyaluronic acid	Aptamer DTA (DEK protein blocker) DSP: Rheumatoid arthritis	Dissolving MNs	The modification of the methoxy group, idT, and cholesterol enhanced the stability and anti-inflammatory efficacy of DTA. The system overcame the skin barrier without pain and quickly released aptamers into the skin.	[33]
Poly (vinyl alcohol)/poly (vinyl pyrrolidone)	Methotrexate DSP: Rheumatoid arthritis	Forming MNs	The integrated patch was able to bypass the skin barrier and deliver MTX in a sustained manner over 24 h. Importantly, the HFMNs were removed intact from the skin with only mild erythema, despite the cytotoxic nature of MTX.	[34]
Liposome hydrogel (LHP)	Triptolide (TP) DSP: Rheumatoid arthritis	Dissolving MNs	When the system was prepared, the hepatic first-pass metabolism and digestive toxicity were eliminated.TP-LHP provided stable, long-term release of triptolide, and had significant efficacy in the CIA model (long-term release 24 h).	[35]
Polyethylene glycol diacrylate (PEGDA)	Gap26 DSP: Keloid	Dissolving MN	Application of Gap 26-loaded MNs could inhibit collagen I expression efficiently, demonstrating the possibility of this peptide loading system in the treatment of keloid scar.	[36]
Poly (vinyl alcohol)/poly (vinyl pyrrolidone	Etravirine DSP: HIV	Dissolving MN	The microneedle formulations exhibited sustained delivery in rats in vivo, suggesting that, following further development, such patches may be clinically useful in that they could replace daily tablets, thus potentially enhancing adherence to therapy and health-related quality of life for HIV patients.	[37]
Poly vinyl pyrrolidone	Rhodamine B, hyaluronidase and millitin DSP: Rheumatoid arthritis	Dissolving MNs	Notably, the permeation-enhancement effect of hyaluronidase (HAase) could be affected by the dosage of HAase and the physicochemical properties of drugs, and therefor formulation optimization is usually required to achieve a satisfying result. Moreover, the interaction between Haase and MN polymers or APIs, like electrostatic force and hydrogen, may prolong the release of HAase and APIs from MNs, and further studies could be performed to illuminate how the formulation components affect the x’ release behavior and bioavailability of HAase MNs.	[38]
WOUND HEALING				
Chitosan	Centella asiática (medicinal herb) DSP: Wound healing	Dissolving MNs	The system was biocompatible with keratinocytes and fibroblasts and showed a sustained drug release over 48 h.	[39]
Chitosan/fucoidan nanoparticle-loaded pullulan microneedle	Moxifloxacin (MOX), thrombin (TH) and lidocaine (LH) DSP: Wound healing	Dissolving MNs	Rapid release of thrombin (TH) and lidocaine (LH) within 1 h, and sustained release of MOX for 24 h. The system heals mice skin wounds completely within 7 days and restores collagen deposition with accelerated cell proliferation, granulation, and reduced proinflammatory cytokines.	[40]
Polyvinyl alcohol	Parathyroid hormone (PTH) DSP: Wound healing	Forming MNs	demonstrated an intermittent systemic administration of PTH using our PTMN patches that accelerated skin wound healing, evaluated for 14 days.	[41]
Hyaluronic acid	Endothelial growth factor (VEGF) DSP: Wound healing	Dissolving MNs	The system could effectively deliver active substances and demonstrated induction of the orientation of fibroblasts; while VEGF release could facilitate tubular formation of endothelial cells.	[42]
Alginate/PEGDA hydrogel 3D-printed hydrogel-filled microneedle	Rhodamine B/bovine serum albumin (BSA) and VEGF DSP: Wound healing	Hollow MNs	The system was effective at containing and releasing bioactive VEGF and promoting gap closure in an HUVEC scratch assay.	[43]
Gelatin and methacrylic anhydride	Adeno-associated virus (AAV) expressing human VEGF (AAV-VEGF) DSP: Wound healing	Dissolving MNs	The implantation did not elicit an obvious inflammatory response and had good biocompatibility in the brain. In addition, the system increased VEGF expression and enhanced functional angiogenesis and neurogenesis.	[44]
Mussel adhesive protein (MAP)-based shell and a nonswellable silk fibroin (SF)-based core	Fluorescein isothiocyanate (FITC)-conjugated dextran DSP: Wound healing	Forming MN	The protein-based hydrogel system achieved in vitro sustained releases for at least 7 days via swelling-mediated diffusion and enzymatic degradation.	[45]
METABOLIC DISORDERS				
CaCO_3_ microparticles (INS-CaCO_3_ MPs) and poly vinyl pyrrolidone (PVP)	Insulin DSP: Diabetes	Dissolving MNs	This study suggests that the use of INS-CaCO_3_/poly vinyl pyrrolidone MNs achieved both high efficiency and constant release of insulin in comparison with the traditional subcutaneous injection approach.	[46]
Alginate with hydroxyapatite biolinker glucose responsive	Insulin DSP: Diabetes	Dissolving MNs	The microneedle patches regulated the blood glucose levels of diabetic mice in normoglycemic ranges for up to 40 h and alleviated the diabetic symptoms of the mice.	[47]
Chitosan	Insulin DSP: Diabetes	Coated MNs	In a type 1 diabetic mouse model, application of the smart MN system quickly causes normoglycemia within 1 h, which is prolonged for 5 h.	[48]
Poly-c-glutamic acid (c-PGA) and poly vinyl pyrrolidone-polyvinyl alcohol (PVP/PVA)	Insulin DSP: Diabetes	Dissolving MNs	The MNs patch after insertion dissolved into the skin within 4 min to deliver the entire drug load, without requiring the users to remove any sharps or waste.	[49]
Polyethylene glycol, 2-methoxyethanol, glycidyl methacrylate, Trimethylolpropane trimethacrylate, and triethylene glycol dimethacrylate (TEGDMA)	Insulin DSP: Diabetes	Dissolving MNs	The system exhibited excellent skin penetration ability and good biocompatibility without skin irritation and hypersensitivity. In vivo transdermal delivery of insulin nanovesicles in diabetic rats demonstrated that the system coupled with iontophoresis could effectively regulate blood glucose levels, maintain normoglycemia, and avoid the critical risk of hypoglycemia.	[50]
Gelatin grafted with carboxylic end-capped poly(N-isopropylacrylamide) (PNIPAm)	Insulin DSP: Diabetes	Solid MNs	The system had a temperature-responsive crosslinker for controlled release. Both in vivo and in vitro tests demonstrated the sufficient penetration ability and crosslinking speed of the system for maintaining blood glucose concentrations within the medicable levels for a long period of time.	[51]
Photocrosslinked methacrylated hyaluronic acid (MeHA)	Glucagon DSP: Diabetes treatment problems	Dissolving MN	The system successfully prevented hypoglycemia induced by overdosed insulin administration in rat models. The results from this work warrant further development of the transdermal glucagon delivery system as a solution to potentially life-threatening complications associated with intensive insulin therapy.	[52]
Phenylboronic acid grafted sodium hyaluronate and polyvinyl alcohol	Insulin DSP: Diabetes	Dissolving MNs	In the hypoglycemic experiment of diabetic rats, the microneedle patch effectively pierced the skin and maintained BGLs in the normal range for a long time.	[53]
Multifunctional silk fibroin methacryloyl	Prussian blue nanozymes (PBNs) and vascular endothelial growth factor (VEGF) DSP: Diabetic wound healing	Dissolving MNs	The system exhibits excellent biocompatibility, drug-sustained release, proangiogenesis, antioxidant, and antibacterial properties with sustained release after 9 days.	[54]
Ti_2_C_3_ MXenes-integrated poly-γ-glutamic acid (γ-PGA) hydrogel MNs	Asiaticoside DSP: Diabetic foot ulcer	Dissolving MNs	The system was shown to be a multifunctional subcutaneous drug-delivering system for accelerating diabetic wound healing for 14 days.	[55]
Poly (DMAA-*co*-PyPBA) and PEG_4a_−Diol	Insulin DSP: Diabetes	Dissolving MNs	The release of insulin from the system was accelerated in the presence of glucose. Moreover, short-term blood glucose control in a diabetic rat model following the application of the device to the skin confirms insulin activity and bioavailability.	[1]
NIPPAn and N,N′-Metilenobisacrilamida (MBA) crosslinked 4-(2-acrylamidoethylcarbamoyl)-3-fluorophenylboronic acid (AmECFPBA)	Insulin DSP: Diabetes	Forming MN	Aqueous stability of at least 2 months and sustained performance durability.	[56]
NIPAM, N-Vinylpyrrolidone (NVP), 3-(acrylamido) phenylboronic acid (AAPBA), MBA and photoinitiator were the base of hydrogel	Insulin DSP: Diabetes	Dissolving MNs	The release of insulin on the system surface was uncontrolled by MNs and rapidly finished after ~10 min. However, the release of insulin within MNs is dependent on glucose concentration.	[57]
CENTRAL NERVOUS SYSTEM DISORDERS				
Gantrez^®^ S-97and PEG	Esketamine (ESK) DSP: Depression	Forming MNs	The authors aimed to achieve sustained therapeutic levels in plasma over 24 h using ESK-containing drug reservoirs in combination with hydrogel-forming MNs in this in vivo feasibility study.	[58]
Poly(vinylpyrrolidone) or Gantrez^®^ S-97	Donepezil DSP: Alzheimer’s	Forming MN	The authors aimed to achieve sustained therapeutic levels in plasma over 24 h, using the optimum patch formulation.	[59]
Gelatin methacryloyl (GelMA)	L-DOPA DSP: Parkinson’s	Dissolving MNs	The system released L-DOPA directly entering the blood, which reduces the side effects on the gastrointestinal tract and improves the utilization rate of the drug.	[60]
OCULAR DISORDERS				
NIPAAm-b-poly glutamic acid hydrogel	Dexamethasone DSP: Posterior ocular diseases	Forming MNs	The system was effective in delivering DEX, a hydrophobic drug, improving effective concentrations, and can be considered a promising system for drug delivery in this ocular region.	[61]
INFECTIONS DISORDERS				
PVA and polyvinyl pyrrolidone (PVP)	Albendazole (ABZ) DSP: Parasitic infections	Forming MNs	The overall results showed that this dosage form has a higher potential for transdermal bioavailability and can deliver ABZ more quickly.	[62]
Carbopol^®^ 934 with poly (Ɛ-caprolactone) (PCL)-based nanoparticles (NPs)	Carvacrol (CAR) DSP: Biofilms-infected wounds	Dissolving MNs	Animal studies showed the successful in vivo delivery of CAR-PCL NPs to the skin, where bacteria can grow and form biofilms.	[63]
Carboxymethyl cellulose/polyvinylpyrrolidone	Amphotericin B (AMB) DSP: Cutaneous leishmaniasis	Dissolving MNs	After insertion to the skin, the system was rapidly dissolved to release the encapsulated drug, and the resulted micropores in the skin were quickly resealed within 30 min. Flow cytometry results showed a potent in vitro leishmanicidal activity of AMB-loaded MN patches against the *Leishmania* parasites (up to 86% of the parasites die).	[64]
Gantrez^®^ S-97 and Carbopol^®^ 974P	Cefazolin DSP: Bacterial infection	Forming MN	The system was capable of achieving up to 80 μg Cefazolin delivery into the epidermis within 2 h of application, with the effect lasting for more than 24 h.	[65]
Poly (lactic-*co*-glycolic acid) (PLGA)	Nile red and Amphotericin B DSP: fungal infections	Forming MN	An in vitro release study demonstrated that the release of amphotericin B inside the system would last for a week. An antifungal test revealed the effectiveness of the inserted tips against fungal growth.	[66]
PVA with S-nitrosoglutathione and graphene oxide (GO)	Nitric oxide (NO) DSP: biofilm-infected chronic wounds	Forming MN	The test results showed that an increase in temperature increased the release of NO from the patch and thus enhanced antibacterial performance.	[67]
Gantrez^®^ S-97, PVA and polyvinyl pyrrolidone (PVP)	Rifampicin, Pyrazinamide, Ethambutol dihydrochloride and others DSP: Tuberculosis	Forming MN	The system was able to deliver optimal concentrations of the drugs used in the test. The results of this work demonstrated the versatility of hydrogel formulations to deliver a tuberculosis drug regimen using MNs.	[68]
PVA and polydopamine (PDA) NP	Levofloxacin and alfa-amilase DSP: Bacterial with biofilm infections	Dissolving MN	The system combined antibiotics and local moderate hyperthermia to kill bacteria and eliminate the biofilm in the wound, thereby shortening the period of inflammation to promote wound healing and tissue regeneration.	[69]
Chitosan-NIPAM hydrogel	Vascular endothelial growth factor (VEGF) DSP: Precaution of infections and promotion of wound healing	Dissolving MNs	The system exhibited a superior capability of acceleration in inflammatory inhibition, collagen deposition, angiogenesis, and granulation tissue formation. In addition, a controlled release can be achieved with thermoresponsive NIPAM.	[70]
Chitosan (CS) and zinc nitrate (CS–Zn [II] MNs)	Chitosan and Zn NP DSP: Bacterial biofilm	Dissolving MNs	The system not only can directly transport CS and Zn^2+^ into the bacterial biofilm but also offers a large specific surface area, to facilitate the diffusion within biofilm and effectively eradicate the bacterial biofilm.	[71]
Chitosan polyethylenimine antimicrobial polymer	Amphotericin B DSP: Fungal infections	Dissolving MN	The results of the combination system are attributable to the high bioavailability of therapeutics and synergistic actions of the antifungal polymer and drug.	[72]
Methacrylated hyaluronic acid (MeHA)	Zn-MOF DSP: Bacterial infections and wound healing	Dissolving MNs	The system significantly promotes angiogenesis, deposition of collagen, and reduced inflammation in the wounds.	[73]
PVA and phenylboronic acid	Clindamicyn DSP: Bacterial acne	Dissolving MNs	The drug-loaded MNs are able to penetrate the stratum corneum to improve the drug interaction against *P. acnes*. Meanwhile, the inflammation-mediated sustained drug release continuously keeps a sufficient drug concentration in the therapeutic levels around acne area (treatment of 6 days).	[74]
REPRODUCTIVE SYSTEM DISORDERS				
Methacrylamide hyaluronic acid hydrogel	Collagen type I DSP: Urinary incontinence	Dissolving MNs	The results of experiments showed that the system has good in vitro biocompatibility. In vivo experiments showed that the system could improve the urodynamic index of urinary stress incontinence in mice.	[75]
PVA and polyvinyl pyrrolidone (PVP)	Sildenafil citrate DSP: Erect disfunction	Forming MN	The ex vivo permeation study showed that up to 80% of sildenafil citrate was delivered transdermally from this combined dosage in the evaluation of 14 days.	[76]
Gelatin-methacryloyl (GelMA)	Antioxidant cerium oxide (CeO_2_) nanozyme MNs with stem cell loading DSP: Endometrial repair	Dissolving MNs	The combination in the system offered a flexible and elaborate platform to realize the combination therapy of endometrial injury. The system contributes to the in situ delivery of stem cells at the sites of injury in an efficient and noninvasive manner, possibly promoting the therapeutic effects of stem cells.	[77]
ANALGESIC, ANESTHETIC, AND/OR ANTI-INFLAMMATORY EFFECT				
Poly (methylvinylether/ maelic acid) (PMVE/MA) hydrogel	Caffeine and lidocaine DSP: Analgesic effect	Forming MNs	The system offered a simple and convenient route of drug administration while eliminating the pain associated with the use of hypodermic needles.	[3]
PVA and poly (ethyl- eneimine) with MBA crosslinking agent	Indomethacin DSP: Anti-inflammatory effect	Dissolving MNs	This investigation provided the efficacy of the device for attaining electromodulated drug release in the ex vivo porcine model.	[78]
Chitosan	Meloxican DSP: Anti-inflammatory effect	Dissolving MNs	The good bioavailability through the system demonstrates that chitosan/meloxicam MNs patches may be suitable to manage pain in cattle after routine procedures.	[79]
Gelatin-methacryloyl (GelMA)	Lidocaíne chlorideoride DSP: Anesthetic effect	Dissolving MNs	The system exhibited an analgesic effect for 6 h with good biological safety such that it did not cause irreversible wounds and irritation to the skin, nor did it bring potential side effects.	[80]
Polyvinyl pyrrolidone (PVP)	Sumatriptan succinate DSP: Analgesic effect	Dissolving MN	The calculated diffusion coefficients were one order of magnitude greater than the value estimated when the drug was directly applied to the skin surface. The dissolution rate constant was affected by the concentration of the polymer matrix.	[81]
PVA and maltose (MT)	Sinomenine hydrochloride DSP: Analgesic effect	Dissolving MNs	The in vitro study showed that the permeation rate was nearly stable within 48 h. Pharmacokinetic studies indicated that the system exhibited lower clearance, longer retention time, higher bioavailability, and stability versus SH-loaded hydrogel.	[82]
Polyvinyl pyrrolidone (PVP)	Lidocaíne hydrochloride DSP: Anesthetic effect	Dissolving MN	The system demonstrated sufficient biocompatibility without causing noticeable irritation to the skin. Also, the obtained MNs significantly shortened the onset time of lidocaine hydrochloride generating local anesthesia effects, comparable to creams.	[83]
PVA with poly (ethyleneimine) solution (PEI) and 1 vinylimidazole (VI)	Indomethacin DSP: Anti-inflammatory effect	Dissolving MNs	Both in vitro and in vivo studies revealed a good preliminary indication that the system has electroresponsive capabilities, ultimately facilitating the immediate release of indomethacin.	[84]
Sodium carboxymethyl cellulose (NaCMC) and gelatine A (GEL)	Lidocaine hydrochloride DSP: Anesthetic effect	Dissolving MNs	Long-term release of lidocaine and precaution of side effect in comparison to conventional methods.	[85]
Carboxymethylcellulose and gelatine	Lidocaine hydrochloride DSP: Anesthetic effect	Dissolving MNs	The diffusion experiments revealed a small increase in diffusional permeation when LFS was used in combination with an MN array pretreated skin.	[86]
SKIN LIGHTERING AND SKIN ANTIAGING AGENT				
Polyacrylic acid-co-maleic acid (PAMA) and p PVA	Alpha-arbutin DSP: Skin lightering	Dissolving and forming MNs	α-arbutin loaded in dissolving MNs had significantly enhanced α-arbutin permeation both in vitro and in vivo more than α-arbutin-loaded forming MNs. In a long-term stability test, α-arbutin remained stable at 25 °C for 3 months.	[87]
1,4-butanediol diglycidyl ether (BDDE) crosslinked hyaluronic acid	Sulforhodamine B. DSP: Antiaging	Dissolving MNs	The system has prolonged effectiveness, with high swelling retention time or epidermal expansion time in mice up to 6 days or more.	[88]
PVA and carbomer	Glabridin DSP: Antiaging	Forming MN	The in vitro release studies showed that cumulative permeation amount within 24 h of the system with chemical crosslinker was significantly higher than that achieved by the system with physical crosslinker and that achieved by glabridin-loaded gel.	[89]
TISSUE REGENERATION				
Gelatin methacryloyl (GelMA) MNs with and without poly(lactic-coglycolic acid) nanoparticles and cytokine-loaded silica microparticles	Tetracycline and cytokines (IL-4 and TGF-b) DSP: Periodontal tissue regeneration	Dissolving MNs	In vivo delivery of the MN patch into periodontal tissues suppressed proinflammatory factors and promoted proregenerative signals and tissue healing.	[90]
Gelatin methacryloyl	Galunisertib DSP: Myocadiac infarction	Dissolving MNs	The drug-release properties can be controlled by adjusting the degree of crosslinking to ensure sustained release for at least 15 days after application.	[7]
MANAGEMENT OF PHARMACOKINETIC AND TOXICITY				
Poly (methylvinylether-co-maleic acid) crosslinked by esterification with poly (ethylene glycol)	Metformin hydrochloride DSP: Enhance bioavailability	Forming MNs	Metformin HCl plasma concentration was shown to be sustained over 24 h. This may indicate that the drug continues to be released from the MN patch while it is being cleared from the body of the rats.	[91]
Polyacrylic acid	Naloxone DSP: Management of side effects of opioids	Dissolving MNs	Naloxone permeation through intact skin was highest from pH 7.4 gels when naloxone is unionized.	[92]
PULMONARY DISORDER				
PVA and polyvinyl pyrrolidone (PVP)	Sildenafil citrate DSP: Pulmonary hypertension	Forming MN	Increasing SC bioavailability in treating pulmonary arterial hypertension is another benefit of this preparation.	[93]
NOT SPECIFIED				
Phenylborate ester bonds crosslinking NIPAAm/ poly(butyl acrylate) PBA	Rhodamine B drug DSP: Not specified	Dissolving MNs	The thermo and glucose-responsive system presented excellent biocompatibility for drug delivery at 37 °C.	[94]
Polyvinylpyrrolidone-based graphene oxide HG	L-Ascorbic acid DSP: Not specified	Dissolving MNs	New platform electricity responsive for drug delivery. The drug-release efficiency of the system with graphene oxide was increased by more than 2 times compared to the system without graphene oxide when 5 V was applied.	[95]
Poly (vinyl alcohol)/poly(vinylpyrrolidone)/citric acid composite crosslinked	Drug models theophylline (THEO), cyanocobalamin (CYAN), and fluorescein sodium (FLUO) DSP: Not specified	Forming MNs	The hydrogel shows good permeability with sufficient mechanical strength to aid skin insertion and hydrophilic drug retention capacity.	[96]
Alginate hydrogel	Ascorbic acid and tranexamic acid DSP: Not specified	Forming MN	The in vitro transdermal delivery efficiencies of drugs were significantly improved throughout 16 h.	[97]
PVP with gold and silver electrodes	Ascorbic acid DSP: Not specified	Dissolving MN	The system was extremely efficient in controlling drug release. It was influenced by temperature, irradiation, and electricity.	[98]
PLGA (poly (D, L-lactic co-glycolic acid), PVA, and polyvinyl pyrrolidone (PVP)	FITC Dextran DSP: Not specified	Dissolving MNs	The cumulative permeation of FITC-Dextran after 48 h was 2.34 ± 0.40 μg/cm^2^ and it showed a 31-fold higher permeation profile than its respective control. This study demonstrated that incorporating higher-molecular-weight molecules into PLGA MNs proved to be an effective strategy to sustain the release of macromolecules across the stratum corneum/epidermis for transdermal delivery.	[99]
PVA	Rhodamine B DSP: Not specified	Dissolving MN	In this study, the system showed dissolution after 17 min, good swelling properties, and a consistent release of the model drug RD for up to 12 days.	[100]
Gantrez^®^ S-97 and PVA with crosslinker polyethylene glycol (PEG) and anhydrous sodium carbonate (Na_2_CO_3_)	Amoxicillin (AMX), atenolol (ATL), diltiazem (DLT), levodopa (LD), carbidopa (CD), levofloxacin (LVX), and primaquine (PQ) DSP: Not specified	Forming MN	The system was efficient for in vivo delivery of AMX, LD, and LVX at therapeutically relevant concentrations.	[101]
Gelatin methacrylate	Rhodamine B DSP: Not specified	Dissolving MNs	System with effectiveness for rapid release phase and sustained release phase.	[102]
2-hydroxyethyl methacrylate (HEMA) and ethylene glycol dimethacrylate (EGDMA)	Ibuprofen DSP: Not specified	Forming MN	In vitro, this system was able to deliver up to three doses of 50 mg of ibuprofen upon application of an optical trigger over a prolonged period of time (up to 160 h).	[103]
Gantrez-PEG	Caffeine DSP: Not specified	Forming MN	The study demonstrated that the use of microwave radiation significantly reduces the time required for MN preparation.	[104]
Gantrez^®^ S-97 and PEG	Ibuprofen and ovoalbumin DSP: Not specified	Forming MN	In in vitro delivery experiments across excised neonatal porcine skin, approximately 44 mg of the model high dose small molecule drug ibuprofen sodium was delivered in 24 h, equating to 37% of the loading in the lyophilized reservoir.	[105]
Poloxamer hydrogel (pluronic)	Methotrexate DSP: Not specified	Dissolving MN	The system embedded within the microporated skin site provided a steady and sustained delivery of methotrexate for 72 h.	[106]
Zwitterionic sulfobetaine (SPB) monomer blended with dextran-glycidyl methacrylate/acrylic acid	Rhodamine B DSP: Not specified	Coated MN	The hydrogel MNs with SPB side chains are suitable for biopharmaceutical transdermal drug delivery, particularly protein-based drugs with improved bioavailability. The MNs exhibited high drug loading capacity, efficient drug release, and protein preservation by suppressing aggregation.	[107]
Cyclodextrin (CD)	Ibuprofen DSP: Not specified	Dissolving MN	In this study, the system cannot only load hydrophilic drugs but also encapsulate hydrophobic drugs through the unique supramolecular structure of cyclodextrin.	[108]
Methacrylated hyaluronic acid (MeHA)	Doxorrubicin and others DSP: Not specified	Coated MNs	In this work, a self-adhesive MN platform with universal drug-loading capability was developed.	[109]
pNIPAM	Insulin DSP: Not specified	Hollow MNs	The system exhibited an evident volume shrinkage, excellent photothermal ability, and repeatable NIR-responsiveness.	[110]
Poly (ethylene glycol) diacrylate/polyvinylpyrrolidone (PEGDA/PVP)	Rhodamine B DSP: Not specified	Dissolving MNs	The system presented cumulative drug release kinetics in 172 h.	[111]
Chitosan-porous carbon nanocomposites	Cephalexin DSP: Not specified	Dissolving MNs	The system responded to acidic pH 4 and electric pulse 5 V for drug release.	[112]
Carboxymethylcellulose	Diclofenac sodium DSP: Not specified	Dissolving MN	The dissolution of the patches in saline buffer results in a maximum cumulative release of 98% of diclofenac after 40 min, and insertion in a skin simulant reveals that all MNs completely dissolve within 10 min.	[113]

Description. Defined specific proposal (DSP): a reference to the applicability defined during the inspection of the articles. When there was no evidence of possible applicability during reading, it was defined as not specified.

## Data Availability

Data sharing not applicable.

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
