# Peer review of "Hydrogel-Based Microneedle as a Drug Delivery System"

_pharmaceutics, 2023, doi:10.3390/pharmaceutics15102444_

Round 1

Reviewer 1 Report

The abstract needs to be improved namely reference to the terms and database used in the systematic review.

Methods: The literature search and selection method followed the Preferred Reporting Items for Systematic Review and Meta-Analyses.

The table 1 can be presented in supplementary information ”Table 1 Results according your application, composition, bioactive compound, type and release characteristics for hydrogel system.”

For each analysed and discussed parameter (4.1.1 Stimulus response; 4.1.2 Induced Controlled Release;  4.1.3 Sustained and Long-term therapy; 4.1.4 Localized and Systemic Treatment; 4.1.5 Self Active)  a table referring the supportive papers and the findings found would be better to understand the different parameters discussed.

The Statistical Analysis needs to be revised, which programme was used? Which statistical method? Data are presented without any reference to the numbers of evaluated sample neither the standard deviation found.

Author Response

Dear Reviewer 1, 

We deeply acknowledge the interest demonstrated in our work and the availability to reconsider a revised version of this manuscript.  

Below we provide point-by-point answers to the comments. 
All changes performed to the manuscript are highlighted (yellow color) in the revised version.
We trust that we have carefully and appropriately addressed all the reviewer’s questions and concerns.

Comments and Suggestions for Authors:

  1. The abstract needs to be improved namely reference to the terms and database used in the systematic review.

Response: On page 1, line 22, the abstract was improved.

  1. Methods: The literature search and selection method followed the Preferred Reporting Items for Systematic Review and Meta-Analyses.

Response: Thank you very much for your comments.

  1. The table 1 can be presented in supplementary information ”Table 1 Results according your application, composition, bioactive compound, type and release characteristics for hydrogel system.” 

Response: we believe that table 1 must be in the main document because provides relevant information.

  1. For each analyzed and discussed parameter (4.1.1 Stimulus response; 4.1.2 Induced Controlled Release;  4.1.3 Sustained and Long-term therapy; 4.1.4 Localized and Systemic Treatment; 4.1.5 Self Active)  a table referring the supportive papers and the findings found would be better to understand the different parameters discussed.

Response: In supplemental information, a new table with analysed and discussed parameters was added.

  1. The Statistical Analysis needs to be revised, which programme was used? Which statistical method? Data are presented without any reference to the numbers of evaluated sample neither the standard deviation found.

Response: Don`t always systematic reviews include statistical techniques to synthesize the data from several studies at differences of meta-analysis that always must have statistical analysis

Reviewer 2 Report

Dear authors

Many thanks for such an interesting review about a very specific field as microneedles using gels as the basic structure. 

Abundant tables with a lot of references, would be very useful for readers

Best regards

Author Response

Dear Reviewer 2:

We deeply acknowledge the interest demonstrated in our work and the availability to reconsider a revised version of this manuscript.  

Below we provide point-by-point answers to the comments. 
All changes performed to the manuscript are highlighted (yellow color) in the revised version.
We trust that we have carefully and appropriately addressed all the reviewer’s questions and concerns.

Comments and Suggestions for Authors:

Dear authors

Many thanks for such an interesting review about a very specific field as microneedles using gels as the basic structure. 

Abundant tables with a lot of references, would be very useful for readers

Response: Thank you very much for your comments, the manuscript has abundant and big tables a several references.  

Reviewer 3 Report

The manuscript titled- A systematic review of the application and features of hydrogel-based microneedle as a drug delivery system, is comprehensive and well organized. A few major points to consider before accepting this manuscript for publication

1.      Add an expert opinion section before the conclusion to provide the readers with the authors’ view about the field and the future of this field

2.      The discussion section can be elaborated more, discuss more details about the studies that are discussed under each section

3.      Provide a section that discusses the challenges and limitations of these systems

4.      Sentence structure can be improved massively. Too much use of “the” is throughout the manuscript.

5.      Figure 5 and 6 labels should have a bigger font size

6.      Figure 11 needs to be enlarged

Sentence structure needs to be improved

Author Response

Dear Reviewer 3:

We deeply acknowledge the interest demonstrated in our work and the availability to reconsider a revised version of this manuscript.  

Below we provide point-by-point answers to the comments. 
All changes performed to the manuscript are highlighted (yellow color) in the revised version.
We trust that we have carefully and appropriately addressed all the reviewer’s questions and concerns.

Comments and Suggestions for Authors

The manuscript titled- A systematic review of the application and features of hydrogel-based microneedle as a drug delivery system, is comprehensive and well organized. A few major points to consider before accepting this manuscript for publication

  1. Add an expert opinion section before the conclusion to provide the readers with the authors’ view about the field and the future of this field.

Response: the opinion of the authors was added after each section to clarify their aspects on each topic in the manuscript.

  1. The discussion section can be elaborated more, discuss more details about the studies that are discussed under each section

Response: The discussion section was improved.

  1. Provide a section that discusses the challenges and limitations of these systems

Response: a section that discusses the challenges and limitations was added

  1. Sentence structure can be improved massively. Too much use of “the” is throughout the manuscript.

Response: the sentence structure was improved.  

  1. Figure 5 and 6 labels should have a bigger font size

Response: the labels of figures were improved. 

  1. Figure 11 needs to be enlarged.

Response: the figure was enlarged.

Reviewer 4 Report

1. A small description in form of paragraph/table/picture similar to what they have done in MS may be incorporated regarding patents

2.  A small description in form of paragraph/table/picture similar what they have done in MS may be incorporated regarding the marketed products based on this concept to justify the practicality of this drug delivery system

3. The authors overall opinion based on the review of literature on shortcoming and future prospects is highly appreciated

4. The article needs thorough check for typographical and grammatical errors which are plenty. 

The article needs thorough check for typographical and grammatical errors which are plenty.

Author Response

Dear reviewer 4:

We deeply acknowledge the interest demonstrated in our work and the availability to reconsider a revised version of this manuscript.  

Below we provide point-by-point answers to the comments. 
All changes performed to the manuscript are highlighted (yellow color) in the revised version.
We trust that we have carefully and appropriately addressed all the reviewer’s questions and concerns.

Comments and Suggestions for Authors:

  1. A small description in form of paragraph/table/picture similar to what they have done in MS may be incorporated regarding patents.

Response:  We believe that the summarized description of marketed products is enough with respect to patents required by the reviewer.

  1. A small description in form of paragraph/table/picture similar what they have done in MS may be incorporated regarding the marketed products based on this concept to justify the practicality of this drug delivery system.

Response: In section 5.3. a description was added related to marketed products based on this concept to justify the practicality of this drug delivery system.

  1. The authors overall opinion based on the review of literature on shortcoming and future prospects is highly appreciated.

Response: In section 5.3. an overall opinion based on the review of the literature on shortcomings and future prospects was added.

  1. The article needs thorough check for typographical and grammatical errors which are plenty.

Response: the typographical and grammatical errors were corrected across de manuscript.

Comments on the Quality of English Language

The article needs a thorough check for typographical and grammatical errors which are plenty

Response: the typographical and grammatical errors were corrected across de manuscript.

Reviewer 5 Report

The review entitled “A systematic review of the application and features of hydrogel-based microneedle as a drug delivery system” presented nice detailed information on hydrogel-based microneedles and their applicability. However, I have some recommendations for the improvement of the review. Kindly address the following comments:

1.      Typographical and grammatical errors could be observed in the present review. I suggest to rectify all the typo and grammatical error from the review.

2.      Kindly rephrase the (abstract) sentence ‘The pharmacokinetic and pharmacodynamic ………..and other treatment modalities.’ to make it more clear.

3.      Keywords must be in alphabetical order.

4.      Author’s opinion is missing in the review. I suggest authors should add their opinion after each section to clarify their aspects on each topic.

5.      Kindly rephrase the (result) sentence ‘The database search yielded …………. to ± 117 articles following the inclusion criteria’ to make it more clear and remove typographical errors.

6.      The authors have gathered and summarized imperative information pertinent to microneedles application. However, in my opinion, applicability in cancer need to be explored more, if they can add some recent findings particularly related to microneedles applied for cancer from the following references https://doi.org/10.3390/ma15217693; https://pubmed.ncbi.nlm.nih.gov/37294705/;

https://link.springer.com/article/10.1007/s13346-023-01297-9, it will add to the information.

7.       I think there is a typographical error in the entire MS while writing the numbers , (comma) has been used instead of . (dot). Kindly, review and replace.

8.      Add one section on drawbacks, bottlenecks and their solutions pertinent to microneedles applications as a drug delivery system.

Need minor English language and typographical editing.

Author Response

Dear Reviewer 5:

We deeply acknowledge the interest demonstrated in our work and the availability to reconsider a revised version of this manuscript.  

Below we provide point-by-point answers to the comments. 
All changes performed to the manuscript are highlighted (yellow color) in the revised version.
We trust that we have carefully and appropriately addressed all the reviewer’s questions and concerns.

Comments and Suggestions for Authors:

The review entitled “A systematic review of the application and features of hydrogel-based microneedle as a drug delivery system” presented nice detailed information on hydrogel-based microneedles and their applicability. However, I have some recommendations for the improvement of the review. Kindly address the following comments:

  1. Typographical and grammatical errors could be observed in the present review. I suggest to rectify all the typo and grammatical error from the review.

Response: Typographical and grammatical errors were corrected.

  1. Kindly rephrase the (abstract) sentence ‘The pharmacokinetic and pharmacodynamic ………..and other treatment modalities.’ to make it more clear.

       Response: the sentence of the abstract was modified.

  1. Keywords must be in alphabetical order.

       Response: On page 1, line 32, the keywords were reorganized in alphabetical order.

  1. Author’s opinion is missing in the review. I suggest authors should add their opinion after each section to clarify their aspects on each topic.

Response: the opinion of the authors was added after each section to clarify their aspects on each topic in the manuscript.

  1. Kindly rephrase the (result) sentence ‘The database search yielded …………. to ± 117 articles following the inclusion criteria’ to make it more clear and remove typographical errors.

      Response: the sentence was modified.

  1. The authors have gathered and summarized imperative information pertinent to microneedles application. However, in my opinion, applicability in cancer need to be explored more, if they can add some recent findings particularly related to microneedles applied for cancer from the following references https://doi.org/10.3390/ma15217693; https://pubmed.ncbi.nlm.nih.gov/37294705/;

https://link.springer.com/article/10.1007/s13346-023-01297-9, it will add to the information.

      Response: the references suggested by the reviewer were incorporated into the manuscript.

  1. I think there is a typographical error in the entire MS while writing the numbers, (comma) has been used instead of . (dot). Kindly, review and replace.

      Response: The typographical errors across the manuscript were modified.

  1. Add one section on drawbacks, bottlenecks and their solutions pertinent to microneedles applications as a drug delivery system.

      Response: a section on drawbacks, bottlenecks, and their solutions pertinent to microneedle applications as a drug delivery system was incorporated into the manuscript.

Comments on the Quality of English Language

Need minor English language and typographical

Response: the typographical and grammatical errors were corrected across de manuscript.

Round 2

Reviewer 1 Report

I have nothing to add or suggest improvements to this article, it can be considered for publication.

Reviewer 3 Report

Rebuttal Acceptable